# Whole-brain in situ mapping of neuronal activation in *Drosophila* during social behaviors and optogenetic stimulation

**Kiichi Watanabe[1†, ‡], Hui Chiu[1§], David J Anderson[1,2]\***

[1]Division of Biology and Biological Engineering, Tianqiao and Chrissy Chen Institute for Neuroscience, California Institute of Technology, Pasadena, United States; [2]Howard Hughes Medical Institute, Chevy Chase, United States

**\*For correspondence:**
wuwei@caltech.edu

**Present address:** [†]International Center for Cell and Gene Therapy, Fujita Health University, Toyoake, Japan; [‡]Department of Medical Research for Intractable Disease, Fujita Health University, Toyoake, Japan; [§]Department of Immunobiology, Yale University School of Medicine, New Haven, United States

**Competing interest:** The authors declare that no competing interests exist.

## eLife Assessment

This work reports an **important** new method for activity-dependent neuronal labeling in *Drosophila* using in situ hybridization, with the potential to establish a new standard in the field. The authors demonstrate the method's applicability by generating **compelling** evidence of the function of male-specific neurons in both aggression and courtship behaviors. These results and the new method will be of great interest to the neuroscience community.

**Abstract** Monitoring neuronal activity at single-cell resolution in freely moving *Drosophila* engaged in social behaviors is challenging because of their small size and lack of transparency. Extant methods, such as Flyception, are highly invasive. Whole-brain calcium imaging in head-fixed, walking flies is feasible but the animals cannot perform the consummatory phases of social behaviors like aggression or mating under these conditions. This has left open the fundamental question of whether neurons identified as functionally important for such behaviors using loss- or gain-of-function screens are actually active during the natural performance of such behaviors, and if so during which phase(s). Here, we perform brain-wide mapping of active cells expressing the Immediate Early Gene *hr38* using a high-sensitivity/low background fluorescence in situ hybridization (FISH) amplification method called HCR-3.0. Using double-labeling for hr38 mRNA and for GFP, we describe the activity of several classes of aggression-promoting neurons during courtship and aggression, including P1[a] cells, an intensively studied population of male-specific interneurons. Using HI-FISH in combination with optogenetic activation of aggression-promoting neurons (opto-HI-FISH), we identify candidate downstream functional targets of these cells in a brain-wide, unbiased manner. Finally, we compare the activity of P1[a] neurons during sequential performance of courtship and aggression, using intronic vs. exonic *hr38* probes to differentiate newly synthesized nuclear transcripts from cytoplasmic transcripts synthesized at an earlier time. These data provide evidence suggesting that different subsets of P1[a] neurons may be active during courtship vs. aggression. HI-FISH and associated methods may help to fill an important lacuna in the armamentarium of tools for neural circuit analysis in *Drosophila*.

## Introduction

Monitoring neuronal activity during specific actions or activities is critical to understanding how the brain controls behavior. Measurements within a local brain region of neural activity in non-transparent awake, behaving animals such as mice or fruit flies can be achieved with electrophysiological and calcium imaging techniques (*Buzsáki, 2004*; *Grewe and Helmchen, 2009*). In *Drosophila*, two-photon

calcium imaging (TPI) has been widely used (*Seelig et al., 2010*) because single-unit electrical recording in the fly central brain remains exceedingly difficult (*Turner et al., 2008*; *Wilson et al., 2004*). But TPI in *Drosophila* requires head-fixation, which limits the repertoire of behaviors that flies can perform under these conditions. For example, it is difficult to use TPI to image neuronal activity during the consummatory phases of complex social behaviors, such as male–male aggression or male–female courtship (but see *Clowney et al., 2015*). Recently, a new method called Flyception2 was introduced for brain imaging in freely walking flies (*Grover et al., 2020*; *Grover et al., 2016*). However, the spatial resolution of this method is limited, it is invasive, and monitoring the activity of single neurons in freely moving animals is still challenging.

In mice, immediate early genes (IEGs) such as c-fos, Arc, or Egr1 have been used to map the activity of neurons by visualizing their expression histologically using immunohistochemistry (IHC) or in situ hybridization (ISH) (*Morgan and Curran, 1991*). More recently, this method has been combined with tissue-clearing and light-sheet microscopy to perform systematic and unbiased whole-brain mapping of neurons activated during different behaviors (*Kim et al., 2015*; *Renier et al., 2016*). An extension of this approach, called <u>c</u>ellular compartment <u>a</u>nalysis of <u>t</u>emporal activity by <u>f</u>luorescence <u>i</u>n <u>s</u>itu <u>h</u>ybridization (catFISH), enables within-animal comparisons of neuronal populations activated during two different sequential behaviors (*Guzowski et al., 1999*; *Lin et al., 2011*).

In *Drosophila*, by contrast, the application of IEGs to map whole-brain neuronal activation patterns has been limited. In part, this is because c-*fos* is not a particularly good activity marker in flies (*Chen et al., 2016*). More recently, several studies using fly IEGs such as *Hr38* and *stripe/Egr1* have been reported (*Chen et al., 2016*; *Fujita et al., 2013*; *Takayanagi-Kiya et al., 2023*; *Takayanagi-Kiya and Kiya, 2019*). Nevertheless, this technique has been relatively under-utilitized in flies. This may be the case, at least in part, because application of FISH to the adult *Drosophila* brain has been technically challenging, due to limited probe penetration and high background caused by traditional amplification methods such as Tyramide (*Raap et al., 1995*).

To circumvent these problems, we have applied in this study an ISH amplification technique called the hybridization chain reaction (HCR), v.3.0 (*Choi et al., 2018*) to visualize the expression of *Hr38*, an IEG expressed by active neurons in the adult male fly brain (*Fujita et al., 2013*). The design of HCR, which requires conjoint hybridization to target sequences of adjacent pairs of 'half-probes' to achieve amplification, affords both greater sensitivity and specificity, allowing detection of low-abundance transcripts with minimal background noise and off-target hybridization. We call this approach <u>H</u>CR3.0-amplified <u>IEG</u> <u>F</u>luorescent <u>I</u>n <u>S</u>itu <u>H</u>ybridization, or 'HI-FISH' (*Figure 1A*). Although we use *Hr38* as proof-of-concept, this approach is in principle applicable to any IEG.

Here, we demonstrate three different applications of HI-FISH relevant to understanding the neural circuitry underlying social behaviors in flies. First, we have asked whether specific neuronal populations discovered in functional screens for aggression-promoting neurons (*Asahina et al., 2014*; *Hoopfer et al., 2015*; *Watanabe et al., 2017*) are indeed active during natural aggressive behavior, courtship, or both. We have further investigated whether such activity is only observed during the contact-mediated consummatory phase of these behaviors, or can also be detected in freely moving flies in the absence of fighting or mating. Second, we have performed unbiased, whole-brain functional mapping of the downstream targets activated by optogenetic stimulation of a neuronal population of interest, a variant method we call 'opto-HI-FISH'. We have applied this method to P1ᵃ neurons (*Anderson, 2016*; *Clowney et al., 2015*; *Hoopfer et al., 2015*; *Inagaki et al., 2014*) a subset of the male-specific P1 class of interneurons originally identified based on their ability to trigger male courtship behavior when stimulated (*von Philipsborn et al., 2011*). Finally, we have developed a variant of the catFISH technique (*Guzowski et al., 1999*), using dual-color FISH combining intronic and exonic *Hr38* probes, to compare neuronal populations activated during two different sequential behaviors, a method we call 'HI-catFISH'. Since experimental activation of P1ᵃ neurons can promote both aggression and courtship in pairs of male flies (*Hoopfer et al., 2015*), we have used HI-catFISH to investigate whether separate or overlapping subpopulations of P1ᵃ neurons are active during natural occurrences of these two social behaviors. Together, the results of these studies illustrate the utility of HI-FISH, opto-HI-FISH, and HI-catFISH to detect the activation of both known and previously unidentified neurons during naturalistic social behaviors that are not amenable to head-fixation. The results have also yielded new biological insights into the neural circuit-level control of aggression and courtship.

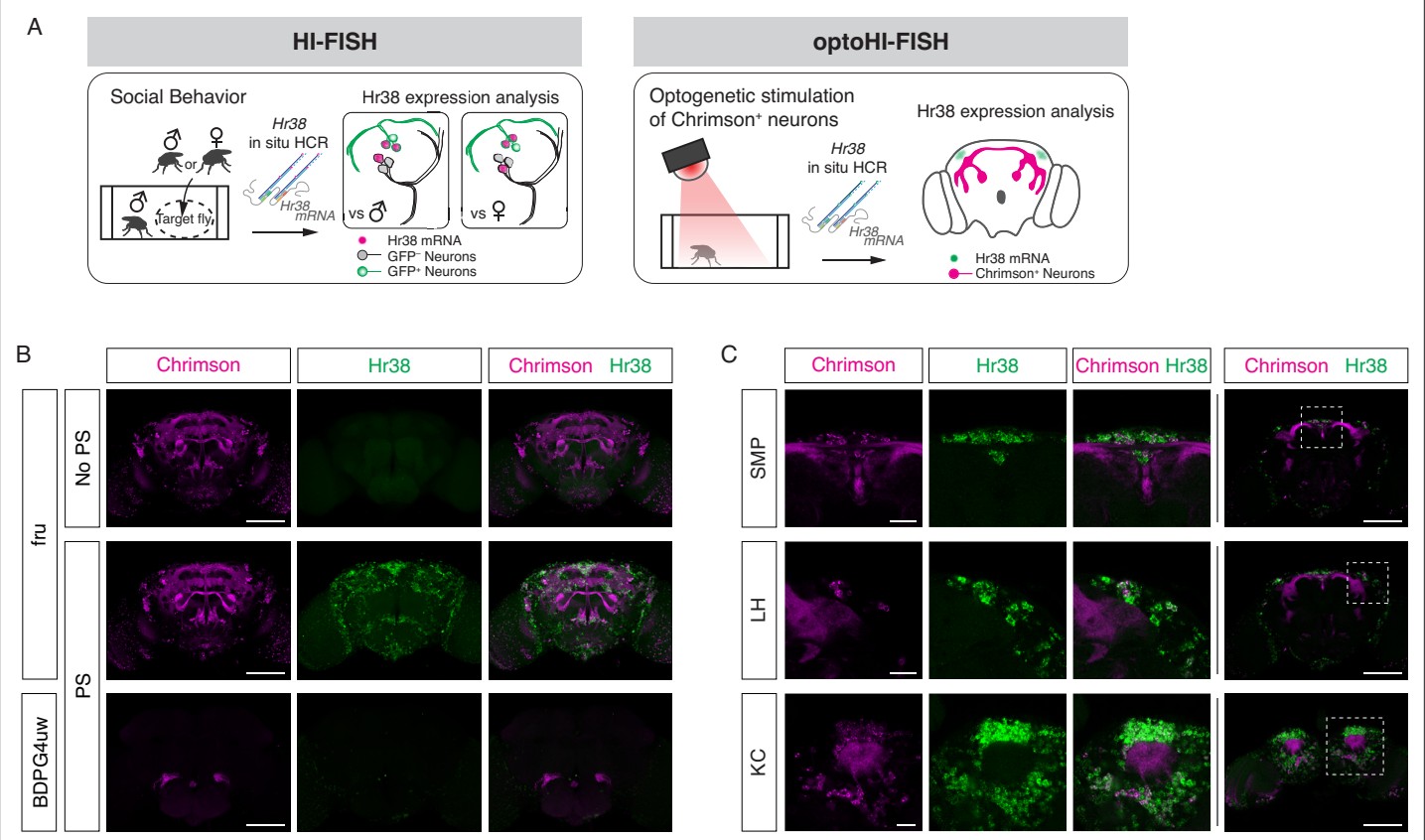

**Figure 1.** Mapping neuronal activities using Hr38 fluorescence in situ hybridization (FISH). (**A**) Illustrated summary of this study for mapping Hr38 expression induced by social behavior or artificial activation of specific neuronal populations. (**B**) Representative images of Hr38 expression before (fru-No PS) and after photostimulation of fru-GAL4>Chrimson neurons (fru-PS). BDPG4uw-PS represents the images after photostimulation of empty-GAL4>Chrimson neurons. magenta: Chrimson::tdT (native fluorescence), green: Hr38 hybridization chain reaction (HCR) signals. Scale bar: 100 μm. (**C**) Magnified confocal section images of Hr38 expression (left three columns) from the boxed region (far right column) after photostimulation of fru-GAL4>Chrimson neurons. SMP, superior medial protocerebrum; LH, lateral horn; KC, Kenyon cells. Scale bar: 20 μm (left three columns) or 100 μm (far right column).

The online version of this article includes the following figure supplement(s) for figure 1:

**Figure supplement 1.** Hybridization chain reaction (HCR) fluorescence in situ hybridization (FISH) and Hr38 signals in the adult fly brain.

## Results

### Brain-wide expression of *Hr38* detected by HI-FISH

The HCR3.0 method provides greater sensitivity, specificity, and lower background than traditional FISH using Tyramide signal amplification. Application of older FISH methods in the *Drosophila* adult whole brain has been limited, due to low sensitivity and high background (*Wilkie and Davis, 1998*). For example, the detection by FISH of low-abundance transcripts such as those encoding G protein-coupled receptors (GPCRs) has been virtually impossible in adult flies up to now. Application of the HCR-FISH method, by contrast, was able to detect mRNA encoding an octopamine receptor Oct-TyrR (*Arakawa et al., 1990*; *Saudou et al., 1990*), as well as Dh44, a high-abundance neuropeptide-encoding transcript (*Figure 1—figure supplement 1A, B*; *Choi et al., 2018*). Background in controls lacking the HCR initiator probes was essentially undetectable.

Next, we investigated the ability of HCR to detect brain-wide expression of *Hr38* following artificial (optogenetic) activation of fruitless-GAL4 (*Stockinger et al., 2005*) neurons expressing the red-shifted opsin Chrimson::tdT. Fru-GAL4 labels ~2000 neurons in the adult male brain, providing a large number of potentially activatable cells. Following photostimulation, *Hr38* signals were detected in virtually all Chrimson::tdT$^+$ neurons as well as in other tdT$^-$ neurons (*Figure 1B, C* and *Figure 1—figure supplement 1C*). The latter may be indirectly activated via synaptic inputs from Fru-GAL4$^+$

neurons. Control flies of the same genotype but without photostimulation, or photostimulated flies with an 'empty' GAL4 driver (BDPG4uw; *Pfeiffer et al., 2008*) displayed much lower levels of *Hr38* expression.

While performing these experiments, we observed that simply transferring flies to a new arena induced a significant level of *Hr38* expression throughout the brain (*Figure 1—figure supplement 1D, E*), likely reflecting the stress caused by the manipulation and new environment. Based on this observation, we conducted subsequent experiments after flies were habituated to the testing arena before starting the assays (see Methods). This minor modification proved essential to distinguishing behavior-specific HI-FISH signals from the widespread *Hr38* induction caused by transfer of the flies from their home vial to an experimental apparatus.

## Mapping of neurons activated by social behaviors

As an initial proof-of-concept application of HI-FISH, we asked whether neuronal subsets initially identified in functional screens for aggression-promoting neurons (*Asahina et al., 2014*; *Hoopfer et al., 2015*; *Watanabe et al., 2017*) were actually active during natural aggressive behavior. These included P1ᵃ (*Hoopfer et al., 2015*), Tachykinin-FruM⁺ (Tk^FruM) (*Asahina et al., 2014*), and aSP2 neurons (*Watanabe et al., 2017*). Despite the fact that activity in these neuron subsets was shown to be essential for normal aggression, it was formally possible that the behavioral phenotypes caused by their artificial stimulation were due to their supra-physiological activation. Therefore, we performed HI-FISH on fly lines carrying a myr::GFP reporter in the cell populations of interest in males paired with different types of target flies under different conditions. Thirty to 45 min after a 20-min interaction with the target fly, we fixed the brains and performed double-label HI-FISH together with fluorescent antibody staining of myr::GFP-expressing neurons (*Figure 1A*; see Methods).

In the control condition, where the experimental flies were kept alone during the observation period, there was little or no detectable *Hr38* expression in P1ᵃ, Tk, or aSP2 neurons (*Figure 2A–D*, 'Control'). In contrast, we observed clear *Hr38* signals in GFP⁺ P1ᵃ, Tk, and aSP2 neurons after a male–male interaction that included fighting behavior (Aggression: *Figure 2A–D*, *Figure 2—figure supplement 1*). We also observed P1ᵃ-negative Hr38⁺ neurons neighboring P1ᵃ neurons during aggression (*Figure 2*, P1ᵃ Hr38). These cells may correspond to other subpopulations of P1 neurons, or to Dsx-expressing pC1 neurons that have also been implicated in aggression (*Koganezawa et al., 2016*). Note that images are single confocal sections provided for qualitative, illustrative purposes only and should not be interpreted quantitatively. Our conclusions are based on statistically analyzed quantitative data presented in bar and box plots. In the case of courtship with a female, comparably strong *Hr38* induction was observed in P1ᵃ and aSP2 neurons, while there was no *Hr38* expression in Tk neurons (*Figure 2A–D*, *Figure 2—figure supplement 1*; Courtship). These data are consistent with prior observations indicating that the functional manipulation of P1ᵃ or aSP2 neurons in the male flies affects both male–male aggression and male–female courtship behavior (*Hoopfer et al., 2015*; *Takayanagi-Kiya and Kiya, 2019*; *Watanabe et al., 2017*), while the manipulation of Tk neurons affects only aggression (*Asahina et al., 2014*).

Social interactions in flies consist of multiple phases: approach, investigation, initial contact, and progression to consummatory behaviors (*Chiu et al., 2021*; *Yamamoto and Koganezawa, 2013*). While electrophysiological recording or calcium imaging can identify neuronal activation that is time-locked to specific behavioral actions, the detection of *Hr38* (or any IEG) induction may reflect activation at any or all of these sequential phases due to the long-lived, accumulated expression of the IEG mRNA. To identify the phase(s) of social behavior wherein *Hr38* induction occurred, we separated the signals or interactions characteristic of these sequential phases in space rather than time. First, we asked whether the activation of these neurons during a male–male social interaction requires actual fighting behavior, or simply close contact with the target male. Initially, we paired the experimental male fly with a dead male, as a source of chemosensory cues. Because some male-derived cues (e.g., visual motion) may require a live conspecific, we also placed the experimental and tester males in an arena with an agarose substrate. Under these conditions, physical contact but not fighting occurs (*Lim et al., 2014*). In both 'touching but no fighting' conditions, the fraction of Hr38⁺ cells among P1ᵃ, Tk, and aSP2 neurons was slightly lower but not statistically significantly different from that measured during actual aggression, while the intensity of *Hr38* mRNA expression in positive cells was slightly lower in the No Food condition (Dead Male and No Food: *Figure 2B–D*, *Figure 2—figure supplement*

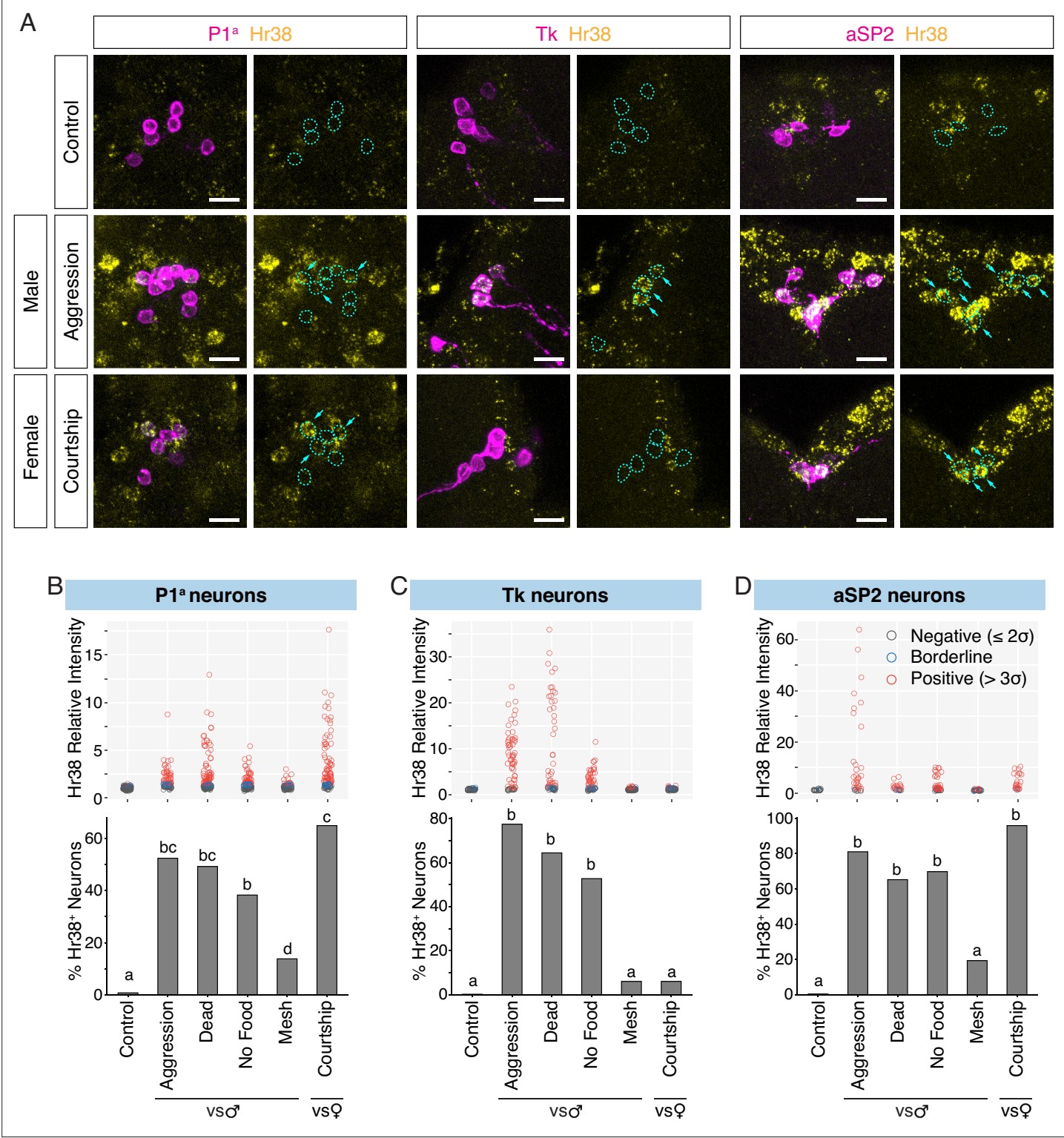

**Figure 2.** Investigation of neurons activated by social behavior with Hr38 fluorescence in situ hybridization (FISH). (**A**) Hr38 expression in the neurons labeled with P1ᵃ-split GAL4, Tk-GAL4, and aSP2-GAL4 drivers. magenta: myr::GFP, yellow: Hr38 hybridization chain reaction (HCR) signals. Dotted lines depicted the outlines of myr::GFP-labeled cell bodies. Arrows: cell bodies of Hr38-positive neurons. Scale bar: 10 μm. *Upper*, scatter plots indicate expression level (relative intensity) of Hr38 HCR signals; *lower*, bar plots indicate percentage of Hr38-positive neurons ('positive' defined as neurons with a relative signal intensity >3σ above the average signal intensity of the control condition) in P1ᵃ (**B**), Tk (**C**), and aSP2 cells (**D**) after different behavioral episodes. Aggression: aggressive behavior against another male, Dead: interaction with a dead male, No Food: interaction with another male without

*Figure 2 continued on next page*

*Figure 2 continued*

food (no fighting), Mesh: interaction with another male, separated with mesh (no physical contact), Courtship: courtship behavior with a virgin female. Bar plots indicate the percentage of Hr38+ neurons among GFP-labeled cells. Bars with the same letter are not statistically significantly different; bars with no common letter are significantly different (p < 0.05, chi-square test). Number of neurons and individuals analyzed; P1ᵃ-Control: 135 (*N* = 8), P1ᵃ-Aggression: 84 (*N* = 5), P1ᵃ-Dead Male: 126 (*N* = 7), P1ᵃ-No Food: 115 (*N* = 6), P1ᵃ-Mesh: 87 (*N* = 6), P1ᵃ-Courtship: 114 (*N* = 6), Tk-Control: 45 (*N* = 4), Tk-Aggression: 69 (*N* = 6), Tk-Dead Male: 61 (*N* = 6), Tk-No Food: 63 (*N* = 6), Tk-Mesh: 82 (*N* = 8), Tk-Courtship: 66 (*N* = 6), aSP2-Control: 20 (*N* = 3), aSP2-Aggression: 36 (*N* = 4), aSP2-Dead Male: 23 (*N* = 3), aSP2-No Food: 49 (*N* = 4), aSP2-Mesh: 36 (*N* = 4), aSP2-Courtship: 24 (*N* = 3).

The online version of this article includes the following figure supplement(s) for figure 2:

**Figure supplement 1.** Complete dataset of Hr38 expression after social behaviors.

*1*). Thus the activation of P1ᵃ, Tk, or aSP2 neurons does not require consummatory aggressive actions such as lunging, tussling, or boxing (*Chen et al., 2002*).

Interacting male flies can, in principle, exchange visual, auditory, mechanosensory, or chemosensory cues. In the case of chemosensory signals, inter-male aggression has been shown to require both detection of volatile cues by the olfactory system, and of non-volatile cues by the gustatory system (*Fernández and Kravitz, 2013*; *Wang et al., 2011*; *Wang and Anderson, 2010*). While both types of cues are present when an experimental fly is exposed to an intact or a dead male fly, the detection of non-volatile cues requires contact chemosensation (*Vosshall and Stocker, 2007*). We therefore exposed the experimental males to a target fly separated by a mesh filter which prevented physical contact between the freely moving flies. Interestingly, *Hr38* induction in all three cell populations under this condition was significantly weaker than that measured during free interaction with a male on agarose, where contact but no fighting occurred (Mesh: *Figure 2B–D*, *Figure 2—figure supplement 1*). Taken together with the relatively strong *Hr38* induction observed following contact with a dead male (which presumably provides a source of both olfactory and gustatory pheromones), these data suggest that strong activation of P1ᵃ, Tk, and aSP2 neurons requires the integration of both volatile and non-volatile male-derived chemosensory cues. However, the results do not exclude the possibility that these neurons exhibit additional activity during contact-mediated aggression, but which is not statistically detectable by this method in comparison to 'no fighting' controls.

## Detection of *Hr38*+ cells induced by optogenetic stimulation of P1ᵃ neurons in the male brain

Optogenetic and thermogenetic stimulation experiments have shown that P1ᵃ interneurons can promote both male-directed aggression and male- or female-directed courtship (*Hoopfer et al., 2015*; *von Philipsborn et al., 2011*). While the neural circuitry downstream of P1 neurons that promotes female-directed courtship behavior has been well-characterized (*von Philipsborn et al., 2011*), the pathway by which P1ᵃ neurons can promote aggression is poorly understood. As a first step to identify systematically P1ᵃ functional downstream targets, we optogenetically activated P1ᵃ neurons using Chrimson::tdT and mapped the brain-wide pattern of *Hr38* induction in males (*Figure 3A–D*), a method we call 'opto-HI-FISH (*Figure 1A*)'. To analyze these data, images of *Hr38* expression from multiple individuals were registered to a reference brain using antibody labeling of Bruchpilot (Brp), a reference neuropil marker, to generate an average induction pattern (*Figure 3E*; *Cachero et al., 2010*). We then created a voxel-based heat map for the *Hr38* expression from the average image. The heat map indicated that neurons around the posterior dorsal region of the central brain were strongly activated by P1ᵃ stimulation (*Figure 3F*).

To validate this approach, we first asked whether we could detect *Hr38* induction in pCd neurons, which were previously shown by calcium imaging to be (indirect) targets of P1ᵃ neurons (*Jung et al., 2020*). Double-labeling in pCd-GAL4; UAS-GFP flies using HI-FISH and immunostaining for GFP (see Methods) confirmed that *Hr38* was strongly induced in pCd neurons by P1ᵃ photostimulation (*Figure 3G*, pCd). Next, we asked whether we could identify any of the specific cellular targets of P1ᵃ neurons in brain regions that exhibited strong induction of *Hr38*. These regions included Kenyon cells (KCs) and PAM dopaminergic neurons in the mushroom body (*Figure 3F*). We used the GAL4 drivers OK107 (*Connolly et al., 1996*) to label KCs, and 0273 and Ddc-GAL4 to label PAM neurons. Following P1ᵃ optogenetic activation, brains were processed for double-labeling with HI-FISH and anti-GFP antibody staining to identify the cells of interest. We observed evidence of *Hr38* induction in

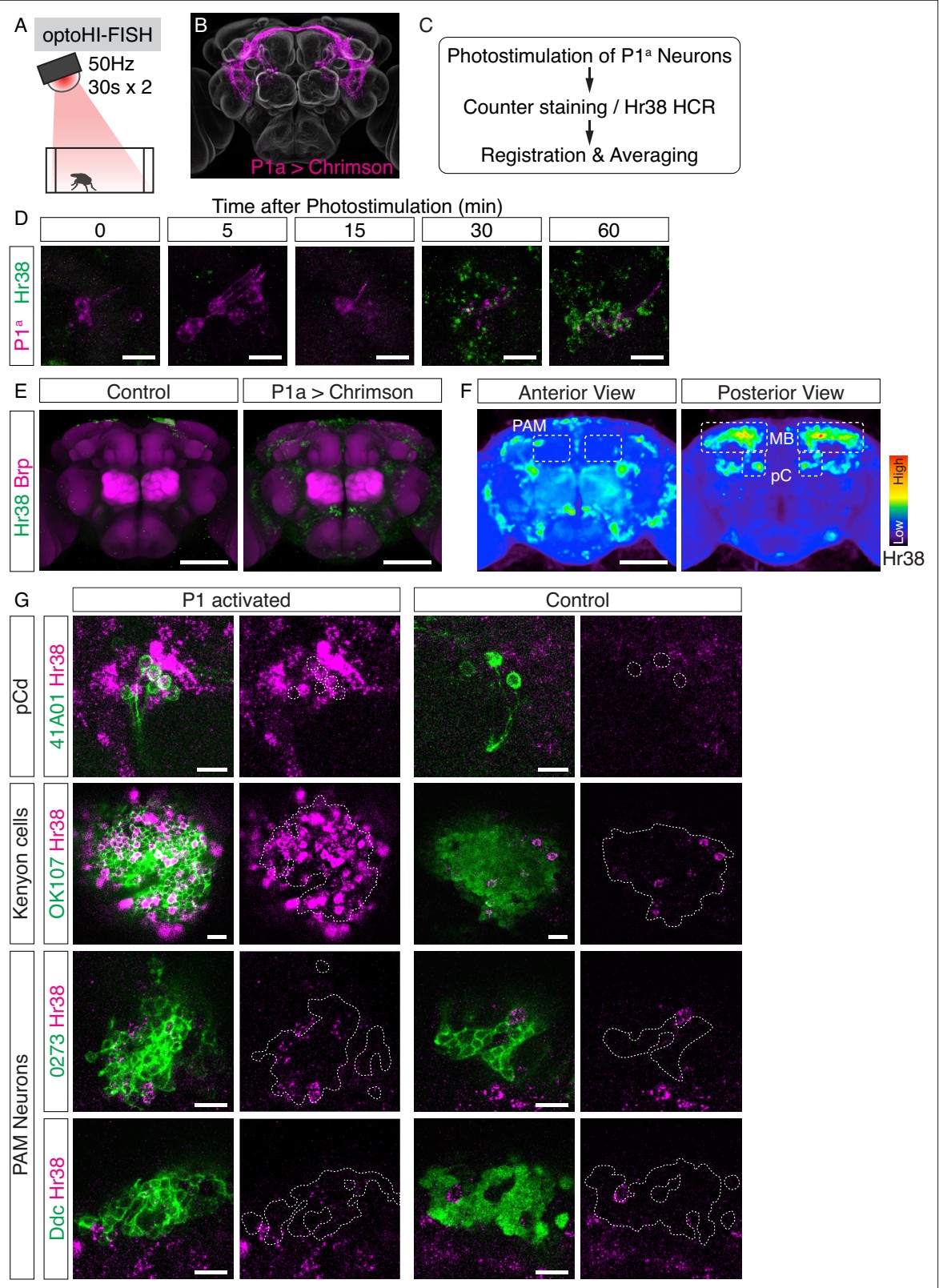

**Figure 3.** Investigation of downstream neurons activated by optogenetic activation of P1ᵃ neurons. (**A**) Illustration of the experimental setup. (**B**) Expression pattern of Chrimson::tdT driven by P1ᵃ split GAL4. (**C**) Scheme of the data analysis procedure. (**D**) Time-course study of Hr38 hybridization chain reaction (HCR) expression after P1ᵃ photostimulation in P1ᵃ-labeled neurons and the surrounding area. By 30–60 min, most of the P1ᵃ-labeled neurons expressed Hr38 (96.8% of P1ᵃ-labeled neurons at 30 min). Scale bars: 10 μm. (**E**) Average image of Hr38 expression of control (BDPG4Uw) and

*Figure 3 continued on next page*

*Figure 3 continued*

P1ª-activated brains. Five individual brains from each condition were registered into a template brain and averaged. Green: Hr38 HCR signals, magenta: Brp. Scale bar: 100 μm. (**F**) Heat map analysis of Hr38 expression. Areas depicted by dotted lines represent the areas of PAM neurons (PAM), mushroom bodies (MB), and pC neurons (pC). Scale bar: 100 μm. (**G**) Confocal images of Hr38 expressions in pCd neurons (R41A01), Kenyon cells (OK107), and PAM neurons (0273 or Ddc) of control and P1ª activated brains. Green: myr::GFP, magenta: Hr38. Dotted lines depicted the outlines of cell bodies (pCd) or cluster of neurons (KC, PAM). Scale bar: 10 μm.

The online version of this article includes the following figure supplement(s) for figure 3:

**Figure supplement 1.** Calcium imaging of PAM neurons and mushroom body neurons combined with P1ª photostimulation.

all three cell populations, although the signal was difficult to quantify due to the high density of GFP+ cells (*Figure 3G*).

To confirm the results of the opto-HI-FISH experiments, we next asked whether PAM neurons and/or KCs were activated in response to P1ª stimulation using in vivo calcium imaging in living, head-fixed flies. We expressed a calcium indicator, jGCaMP7b (*Dana et al., 2019*) in either KCs or PAM neurons using the GAL4/UAS system and Chrimson::tdT in P1ª neurons using the LexA/LexAop system (*Brand and Perrimon, 1993*; *Lai and Lee, 2006*). The neurites of both KCs and PAM neurons showed significant time-locked responses to optogenetic activation of P1ª neurons (*Figure 3—figure supplement 1A–F*). We also imaged jGCaMP7b signals from cell bodies of PAM neurons while photostimulating P1ª neurons, and observed clear responses in both cases as well (*Figure 3—figure supplement 1G–J*). Although we could not identify the specific activated cell populations among the KCs (data not shown), P1ª photostimulation evoked a significant rise in $\Delta F/F$ in~50% of PAM neurons (*Figure 3—figure supplement 1I*). These data validate the results from brain-wide *Hr38* induction mapping and provide proof-of-concept that brain regions containing functional targets of a cell population of interest (here P1ª neurons), as well as specific neuron subtype targets within these regions (KCs and PAM neurons), can be identified by opto-HI-FISH. These results are also consistent with a recent report showing that subsets of PAM dopaminergic neurons receive excitatory inputs from P1 neurons, and assign to KCs a positive valence associated with P1 activation (*Shen et al., 2023*).

## HI-catFISH reveals distinct fighting- and mating-activated P1ª neuron subsets

Previous reports showed that artificial activation of P1ª neurons promoted both aggression and courtship. The foregoing data indicated that *Hr38* was induced in P1ª neurons after both natural aggressive and courtship behavior (*Figure 2*, *Figure 2—figure supplement 1*). An important outstanding question is whether the aggression- and courtship-activated P1ª populations are the same or different (*Anderson, 2016*). Because individual P1 neuron subtypes cannot be distinguished by FISH, between-animal comparisons of *Hr38* induction following fighting or mating cannot distinguish these two possibilities. To compare directly which P1ª neurons are active during naturalistic mating vs. aggression, we developed an *Hr38* analog of catFISH (HI-catFISH) to compare neurons activated during aggression and courtship in the same animal (*Figure 4A*). The catFISH method exploits the different kinetics of nuclear-cytoplasmic transport and of IEG mRNA stability (*Guzowski et al., 1999*) to distinguish neurons activated during a behavior performed just prior to analysis (nuclear pre-mRNA), from those activated during a different behavior performed 30min before the second (cytoplasmic mRNA); cells activated during both behaviors contain both cytoplasmic and nuclear signals (*Figure 4B*). In mice, we have used catFISH with c-*fos* to compare hypothalamic neurons activated during aggression vs. mating in males (*Lin et al., 2011*).

We detected neuronal populations activated during two consecutive social behaviors using HI-FISH containing *Hr38* intronic and exonic probes amplified using different fluorophores. In two color FISH, unspliced pre-mRNAs were detected as nuclear dots labeled by both probes, while mRNA was detected as a more diffuse cytoplasmic signal detected by only the exonic probe (*Figure 4B*). A time-course study using opto-HI-FISH with these two different *Hr38* probes indicated that the primary RNA transcripts first appeared in P1ª nuclei <15 min after optogenetic activation, and disappeared by 60 min. Conversely, cytoplasmic processed mRNAs started to accumulate in the cytoplasm 30 min after activation and disappeared by 2–3 hr after stimulation (*Figure 4—figure supplement 1A–C*).

Having established this calibration of the temporal resolution of HI-catFISH, we compared labeling during free social interactions with a male or a female target. Following pre-habituation (see above),

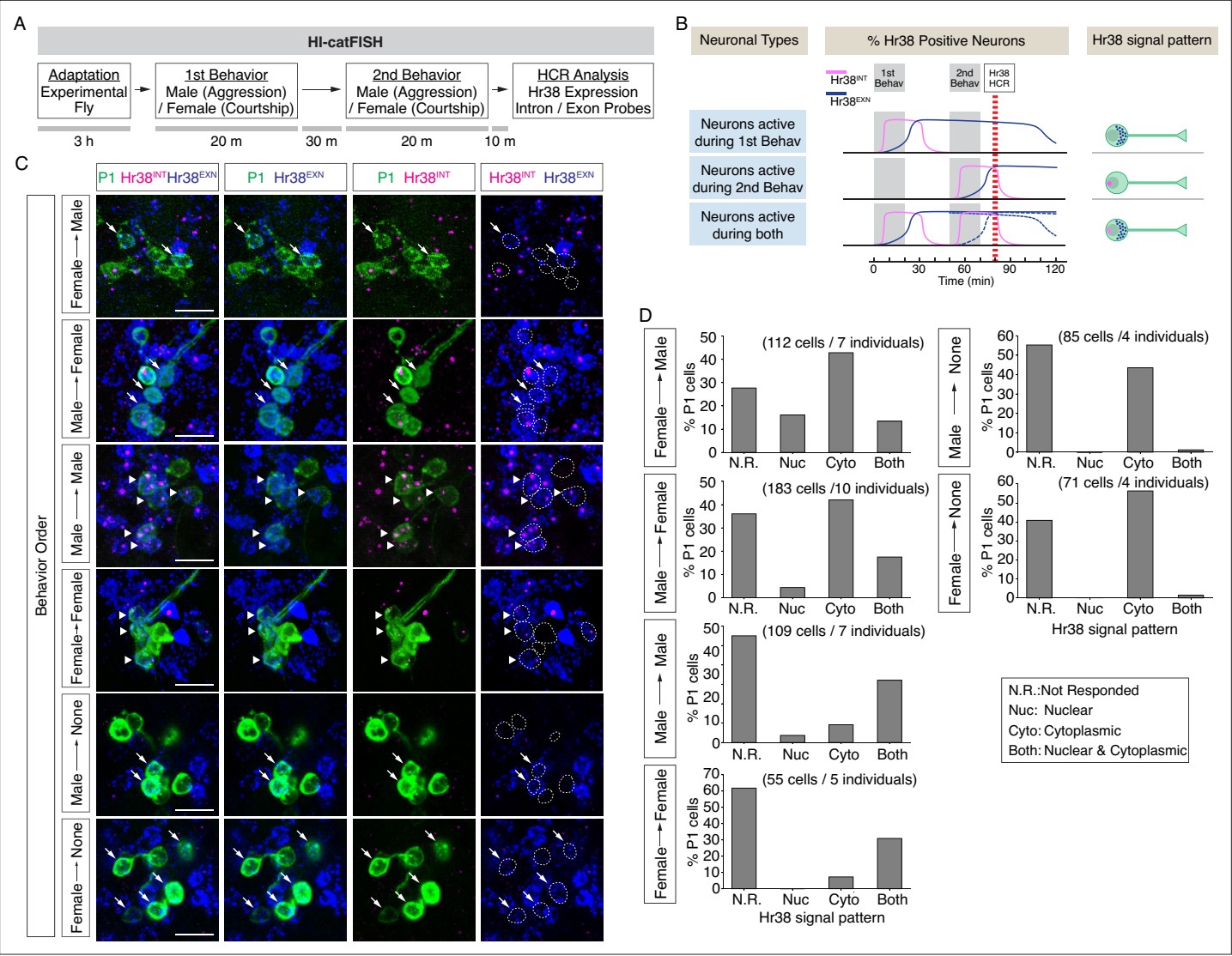

**Figure 4.** *Drosophila* catFISH reveals P1ᵃ subpopulations activated by two different social behaviors. (**A**) Experimental design of the fly catFISH. Each fly experienced the first behavior (aggression or courtship) followed by second behavior (aggression or courtship), and the hybridization chain reaction (HCR) was performed with the Hr38 probes targeted for the exon or the intron sequence. (**B**) Time-course of Hr38 signals detected by the exon- or the intron-targeted Hr38 HCR probes and expected Hr38 signals in each neuron. (Top) The intron-targeted Hr38 HCR signals (magenta) induced by first behavior disappear at the timing of HCR analysis while the exon-targeted Hr38 signals (blue) remain in the cytoplasmic area. (Middle) Both intron- and exon-targeted signals are present in the nucleus. (Bottom) Both intron- and exon-targeted signals induced by the second behavior are present in the nucleus, and exon-targeted Hr38 signals induced by the first behavior are present in the cytoplasmic area. In 'Neurons active during both', the exon probe signals from the first behavior decline (dotted blue line *1) after 60 min (also see *Figure 4—figure supplement 1*). However, it is canceled out as the signals from second behavior are increased (dotted blue line *2). As a result, the combined Hr38 exon probe signals (solid blue line) remain high at the time point for Hr38 HCR. (**C**) Representative images of Hr38 HCR signals in P1ᵃ neurons after six different behavioral conditions. Green: myr::GFP, magenta: intron-targeted Hr38 HCR signals: exon-targeted Hr38 signals. Scale bar: 10 μm. The images shown are for illustration purposes only and represent single optical sections covering only a portion of the P1ᵃ neuronal cluster. They exclude P1ᵃ cells present in other Z-planes. Quantification (**D**) was based on signals measured in all P1ᵃ cells contained within an entire Z-stack. See also *Figure 4—figure supplement 1D*. (**D**) Fraction of responsive P1ᵃ neurons across indicated conditions. N.R.: not responded, Nuc: nuclear signal only (responded to only second target), Cyto: cytoplasmic signal only (responded to only first target), Both: nuclear and cytoplasmic signal (responded to both first and second targets).

The online version of this article includes the following figure supplement(s) for figure 4:

**Figure supplement 1.** Time-course study of the expression of Hr38 with the exon- or intron-targeted probes after photostimulation of P1ᵃ neurons.

a naive experimental male fly was placed into a behavioral arena, and a male or female fly was introduced as a first target. After 20 min of free social interactions the target fly was removed from the arena and, after a further 30-min recovery period, replaced by a second target fly (male or female; Methods). All four combinations of sex of the first and second target fly, and two additional control conditions, in which the experimental flies were exposed to only the first target fly (Male → None, Female → None), were tested (*Figure 4C*). For the two sequential behaviors condition, 10 min after exposure to the second target fly (and~1hr after exposure to the first target fly), the experimental fly was sacrificed and analyzed by dual-color HI-FISH *Hr38* (*Figure 4A, B*).

As expected, many of the neurons activated by the first target were re-activated by the second target (nuclear and cytoplasmic signal), when the sex of the two target flies was the same (*Figure 4C*, Male → Male, Female → Female, arrowheads and *Figure 4D*, 'Both' vs. 'Cyto'). In contrast, few neurons exhibited nuclear Hr38 transcripts detected with the intronic probe when the experimental flies were exposed to the only first target fly but sacrificed after an additional 60min (*Figure 4C*, Male → None, Female → None, 'Nuc' or 'Both'). By contrast, many neurons activated by the first target fly (cytoplasmic signal) were not activated by the second target fly (nuclear signal), when the target sexes were different (*Figure 4C*, Female → Male, Male → Female, arrows and *Figure 4D*, 'Cyto' vs. 'Both'). Responses to males were clearly detected in a subset of P1ᵃ neurons whether the male was presented first (*Figure 4C*, Male → Female; GFP⁺; dashed oval outlines containing blue diffuse cytoplasmic signal) or second (*Figure 4C*, Female → Male; magenta nuclear dots). In addition, there were male-activated cells nearby that were not labeled by the P1ᵃ-split GAL4 driver. In general, these male HI-FISH signals did not overlap with the HI-FISH signals evoked by female targets. We note, however, that relatively few neurons were activated by the female fly when the sex of the two targets was different, regardless of whether the female was presented first or second; the reasons for this are currently unclear but may reflect a suppressive effect of aggression neuron activation on courtship neuron activity, as suggested by earlier studies (*Hoopfer et al., 2015*). These data suggest that there may be distinct P1ᵃ subpopulations that are activated during either male–male aggression or male–female courtship.

## Discussion

*Drosophila* is a powerful model system for identifying and characterizing neural circuits that control innate behaviors. With recent progress in two-photon imaging techniques using genetically encoded calcium sensors, it has become possible to monitor activity from genetically labeled neurons at single-cell resolution in head-fixed flies (*Seelig et al., 2010*). Although single-neuron imaging in unrestrained behaving animals has been achieved in mammalian studies (*Flusberg et al., 2008*; *Remedios et al., 2017*; *Ziv et al., 2013*), most imaging studies with single-cell resolution in *Drosophila* require head-fixed conditions. Therefore, the repertoire of naturalistic behaviors that can be studied using this approach – in particular social behaviors dependent on chemosensory cues (but see *Hindmarsh Sten et al., 2021*)– remains limited.

In mammalian systems, the discovery that IEGs such as c-*fos* are induced by neuronal activity (*Greenberg et al., 1985*; *Greenberg and Ziff, 1984*) has enabled brain-wide functional mapping of neuronal activation following specific behaviors (*Morgan and Curran, 1991*; *Renier et al., 2016*). This method and its derivatives have had a transformative impact on the identification of behaviorally relevant neural circuits in rodents. In *Drosophila*, by contrast, few if any studies describe the application of IEGs for brain-wide activity mapping. In this report, we applied the HCR3.0 RNA-FISH technique (*Choi et al., 2018*) to visualize the expression of a fly IEG, *Hr38* in the entire adult male brain following different manipulations (see also *Shao et al., 2019*). We present data providing proof-of-concept that HI-FISH can be used to detect the activation by naturalistic stimuli of neurons previously identified in thermogenetic behavioral activation screens, as well as to identify candidate downstream targets of these neurons following their optogenetic activation (opto-HI-FISH). We also compared neuronal subsets within a small identified population (P1ᵃ neurons) that is activated during two different social behaviors, using HI-catFISH.

### Monitoring neuronal activities during social behaviors

Previous studies reported that male-specific P1ᵃ, Tk, and aSP2 neurons promote male–male aggression (*Asahina et al., 2014*; *Hoopfer et al., 2015*; *Watanabe et al., 2017*). In addition, P1ᵃ and aSP2

neurons promote male–female courtship (*Hoopfer et al., 2015*; *Takayanagi-Kiya and Kiya, 2019*). Artificial manipulation of the activity of these neurons affected social behaviors and demonstrated their causal roles in these behaviors. However, there was no direct evidence showing that these neurons are normally activated during the specific behaviors they can promote.

Our HI-FISH data demonstrated that P1ᵃ and aSP2 neurons are active during male–male aggression and male–female courtship, consistent with the behavioral effects of activating these neurons (*Figure 2*; *Hoopfer et al., 2015*; *Watanabe et al., 2017*). In contrast, Tk^FruM neurons were only active during male–male aggression (*Figure 2*), consistent with their lack of a functional role in courtship (*Asahina et al., 2014*). Furthermore, we show that P1ᵃ, Tk, and aSP2 neurons can apparently be activated by chemosensory cues from an opponent male fly in the absence of attack behavior (*Figure 2*). These data suggest that P1ᵃ, Tk, and aSP2 neurons may serve to process and integrate sensory cues from the opponent and relay that information to the downstream neurons that encode the consummatory phase of aggression (*Chiu et al., 2021*). However, they do not exclude that these neurons may play a role during overt attack.

Although the time resolution of IEG-based activity monitoring falls far short of that of calcium imaging, our data show that one can nevertheless use HI-FISH to separate activation that occurs during the appetitive vs. consummatory phases of male–male aggression, by judicious manipulation of experimental conditions.

## Identification of downstream target neurons of genetically labeled neurons by opto-HI-FISH

Artificial activation of P1ᵃ neurons induces male–female courtship and an internal state that promotes male–male aggression (*Bath et al., 2014*; *Hoopfer et al., 2015*; *Inagaki et al., 2014*). Furthermore, artificial activation of male-specific P1 neurons mimics courtship-induced place preference as well as supporting appetitive olfactory conditioning (*Shen et al., 2023*). It is critical to identify neurons that are functionally downstream of P1ᵃ cells to understand how these neurons can promote this state. In this study, using opto-HI-FISH, we identified pCd neurons, KCs and PAM neurons as downstream targets of P1ᵃ neurons. These results were confirmed by TPI analysis (*Figure 3*, *Figure 3—figure supplement 1*). Importantly, recent studies have reached the same conclusions regarding neuronal populations using different approaches (*Jung et al., 2020*; *Shen et al., 2023*). These findings provide additional support for our methodology. The heat map illustrates the presence of additional potential P1ᵃ -follower populations beyond the three interrogated here (*Figure 3F*). Further analysis will unveil the detailed characteristics of these potential downstream neurons of P1ᵃ neurons. An important caveat is that not all active *Drosophila* neurons may express *Hr38*. Act-seq data from mice indicate that different neuron subsets activated during social behaviors express different IEGs to different extents (*Kim et al., 2019*).

## Functional heterogeneity of P1ᵃ populations during social behaviors

In addition to identifying functional downstream targets of P1ᵃ neurons, characterizing cellular diversity among P1ᵃ neurons and how this functional heterogeneity maps onto their connectivity with different downstream targets is critical to understanding how these cells can induce both courtship and aggressive behavior. Optogenetic activation of P1ᵃ neurons at a low frequency promotes aggression toward males, while high-frequency stimulation promotes male-directed courtship during stimulation, followed by aggression in the inter-stimulation intervals (*Hoopfer et al., 2015*). P1ᵃ neurons may control these two behaviors according to their activity level and/or the subtype of neuron activated in the population. Alternatively, P1ᵃ neurons may indirectly promote fighting by inhibiting aggression-promoting neurons during courtship (*Koganezawa et al., 2016*); post-inhibitory rebound in such neurons following the offset of P1ᵃ optogenetic stimulation may trigger aggressive behavior (*Anderson, 2016*).

Our preliminary *Hr38* catFISH results suggest the presence of male- or female-responsive subsets within the P1ᵃ population, which consists of 8–10 neurons/hemibrain (*Figure 4*). To our knowledge, this is the first evidence that P1ᵃ neurons are activated during male–male social interactions. Previous studies using calcium imaging of head-fixed flies suggested that P1ᵃ neurons were inhibited by the male-specific pheromone, 11-cis-vaccenyl acetate (*Clowney et al., 2015*). The reason for this apparent discrepancy is not yet clear. The molecular identity of the male- vs. female-responsive P1ᵃ subsets

remains to be determined. It has been suggested that FruM[+] P1 neurons exclusively control courtship, while Dsx[+] P1 neurons (also called pC1 neurons; *Koganezawa et al., 2016*) control aggression. However, triple-intersection experiments have demonstrated that thermogenetic (*Hoopfer et al., 2015*) or optogenetic (*Watanabe et al., 2017*) activation of FruM[+] P1[a] neurons, which comprise only 1–4 neurons/hemibrain, can also promote aggression. Therefore, while our results support the idea that different P1 subsets control aggression vs. courtship, they do not support the view that FruM[+] P1 neurons only control the latter and not the former social behavior.

While this paper was in the final stages of preparation, a manuscript by Takayanagi-Kiya et al. appeared that described a method for labeling neurons activated during innate behaviors using a different IEG, *stripe/egr-1* (*Takayanagi-Kiya et al., 2023*). To analyze neuronal activities in different behavioral contexts, that study utilized a GAL4 driver for the *stripe/egr-1* gene to label neurons activated by the first behavior and immunostaining for stripe protein to label the second behavior. The flies were exposed to the first target for 24hr, followed by exposure to the second target for 2hr. In contrast, our approach for HI-catFISH is based purely on detecting the expression of the IEG mRNA using a state-of-the-art amplification method, HCR v3.0 (*Figure 1—figure supplement 1B*; *Choi et al., 2018*). This enables us to monitor neuronal activity during two different behaviors with much higher temporal resolution than the analogous method of Takayanagi-Kiya. Furthermore, by combining our HI-catFISH with their protein expression-based method, it is theoretically possible to monitor neurons activated by four different behavioral contexts. Thus, the two methods complement each other.

In mice, distinct subpopulations of estrogen receptor 1 (Esr1)-expressing neurons in the ventrolateral subdivision of the ventromedial hypothalamus (VMHvl) are activated during male-directed aggression vs. female-directed mating (*Karigo et al., 2021*; *Remedios et al., 2017*). Our observation that apparently distinct subpopulations of P1[a] neurons are activated during aggression vs. courtship in *Drosophila* strengthens the analogy between these two control nodes for social behavior (*Anderson, 2016*). Application to P1[a] neurons of Act-Seq, a method that uses scRNAseq to identify transcriptomic cell types that are active during different behaviors as determined by IEG expression (*Kim et al., 2019*; *Wu et al., 2017*), may help to identify transcriptomic and functional diversity within the P1[a] population. Such correlative studies may be extended to functional perturbations by using transcriptomic data to identify more specific intersectional GAL4 drivers for distinct P1[a] subsets. Together, our studies illustrate how HI-FISH can provide a systematic, unbiased, and brain-wide approach to identifying entry points to circuits controlling specific social behaviors, which can be integrated eventually with imaging, connectomic, transcriptomic, and functional approaches to circuit analysis.

## Materials and methods

### Fly stocks

Detailed fly genotypes used for each figure are listed in *Supplementary file 1*.

R15A01-iLexA (improved LexA) was generated for this study. Briefly, the R15A01 enhancer fragment (*Jenett et al., 2012*) was cloned into pBPnlsLexA::p65::GADUw (*Chiu et al., 2021*).

20xUAS-IVS-Syn21-Chrimson::tdTomato3.1 (attP2) and 13xLexAop2-IVS-Syn21-Chrimson::tdTomato3.1 (su(Hw)attP5) were obtained from G. Rubin (Janelia Research Campus).

0273-GAL4 (*Burke et al., 2012*) was obtained from S. Waddell (The University of Oxford).

fru[GAL4] (*Stockinger et al., 2005*) was from B. Dickson (Janelia Research Campus).

The following strains were obtained from Bloomington *Drosophila* Stock Center (Indiana University); R15A01-AD (attP40), R71G01-DBD (attP2), R41A01-GAL4 (attP2), OK107, Ddc-GAL4, 20xUAS-IVS-jGCaMP7b (VK00005).

### Optogenetic activation in behaving flies

A detailed description of the setup used for optogenetic activation was described previously (*Inagaki et al., 2014*; *Watanabe et al., 2017*). Flies were raised at 25°C and collected on the day of eclosion and kept in the dark for 7–10 days on the fly media containing 0.4 mM all *trans*-Retinal (Merck, St. Louis, MO). The experiments were performed with the '8-well' chamber (16 mm diameter × 10 mm height) with IR backlight (855 nm, SOBL-6x4-850, SmartVision Lights, Norton Shores, MI). A 655-nm 10-mm Square LED (Luxeon Star LEDs Quadica Developments, Brantford, Canada) was used for

photostimulation. The stimulation protocols for each experiment are described in the figures or figure legends.

## Behavioral assays

Flies were maintained at 9AM:9PM Light:Dark cycle. For male–male aggression assays and male–female courtship assays, the '8-well' acrylic chamber was used as described previously (*Inagaki et al., 2014*; *Watanabe et al., 2017*). The walls of the chamber were coated with a fluoropolymer resin (PTFE-30, Insect-a-Slip) (BioQuip Products, Rancho Dominguez, CA). The clear top ceiling plate was coated with Sigmacote (Merck, St. Louis, MO) to prevent flies from walking on the surfaces. The floor was covered with a layer of apple juice/sucrose-agarose food. The arena was illuminated with IR backlighting.

## Two-photon calcium imaging

Calcium imaging with optogenetic activation was performed as described (*Inagaki et al., 2014*; *Jung et al., 2020*). A deep red (660 nm) fiber-coupled LED (Thorlabs, Newton, NJ) with a band-pass filter (660 nm, Edmund Optics, Barrington, NJ) was used for light source to activate Chrimson.

Flies were collected on the day of eclosion and kept in the dark on the fly media containing 0.4 mM all *trans*-Retinal (Merck, St. Louis, MO). Seven- to ten-day-old flies were cold-anesthetized using a temperature-controlled stage and mounted on a plastic plate with a small window using wax. The cuticle of the dorsal side of the head was removed by forceps. Flies were then placed beneath the objective.

Fly saline (108 mM NaCl, 5 mM KCl, 4 mM NaHCO$_3$, 1 mM NaH$_2$PO$_4$, 5 mM trehalose, 10 mM sucrose, 5 mM N-2-hydroxyethylpiperazine-N-2-ethane sulfonic acid (HEPES), 0.5 mM CaCl$_2$, 2 mM MgCl$_2$, pH = 7.5) was used to bathe the brain during imaging.

## IHC and HCR FISH

In situ HCR was performed as previously described with minor modifications (*Choi et al., 2018*, also see https://www.moleculartechnologies.org/supp/HCRv3_protocol_generic_solution.pdf). For the hybridization buffer, formamide was substituted with an equal volume of 8 M urea solution for better preservation of tissue morphology (*Sinigaglia et al., 2018*).

To process multiple individuals, we collected flies (~8 individuals at once), soaked them in ice-cold ethanol to remove the wax on the cuticle, and opened the head cuticle of each fly with minimal time-lag among samples in the ice-cold fixative (4% paraformaldehyde in phosphate-buffered saline [PBS]). After brief fixation, the brain from each individual was dissected and fixed for 2 hr at 4°C.

For combination of IHC and HCR FISH, we performed IHC as described previously before HCR FISH procedure in RNase-free condition (*Watanabe et al., 2017*). For antibody incubation, we added RNasin Plus (5% of final volume, Promega, Madison, WI). After immunostaining, the samples were post-fixed with 4% formaldehyde in PBS and proceeded for HCR. HCR probes were purchased from Molecular Technologies (https://www.moleculartechnologies.org/). The identifiers for each probe are as follows: Hr38 (exon): Hr38_MT 2627/B655 (B2 Amplifier) and Hr38 3278/C629 (B5 Amplifier), Hr38 (intron): HR38intron 3736/D801 (B1 Amplifier), Dh44: Dh44 3174/C241 (B3 Amplifier), Oct-TyrR: Oct-TyrR_CDS 3008/B815 (B2 Amplifier). We tested two different Hr38 probes (2627/B655 and 3278/C629) and obtained the same results. Images of HCR3.0 data presented in this manuscript were obtained using the Hr38_MT 2627/B655 probe set. The samples were mounted in 88% (wt/vol) Histodenz in 0.02 M Phosphate buffer (pH 7.5 with NaOH) with 0.1% Tween-20 and 0.01% sodium azide (*Yang et al., 2014*).

The antibodies used were as follows: rat anti-DN Cadherin (1:50, DN-EX #8, developed by T. Uemura, obtained from Developmental Studies Hybridoma Bank, University of Iowa, Iowa City, IA), mouse anti-Bruchpilot (1:50, nc82, developed by E. Buchner, obtained from Developmental Studies Hybridoma Bank, University of Iowa, Iowa City, IA), rabbit anti-GFP (1:1000, A11122, Thermo Fisher Scientific, Waltham, MA), Goat anti-Mouse IgG (H+L) Alexa Fluor 488 (1:1000, A-11001, Thermo Fisher Scientific, Waltham, MA), Goat anti-Mouse IgG (H+L) Alexa Fluor 568 (1:1000, A-11004, Thermo Fisher Scientific, Waltham, MA), Goat anti-Mouse IgG (H+L) Alexa Fluor 633 (1:1000, A-21050, Thermo Fisher Scientific, Waltham, MA), Goat anti-Rabbit IgG (H+L) Alexa Fluor 488 (1:1000, A-11008, Thermo Fisher Scientific, Waltham, MA), Goat anti-Rabbit IgG (H+L) Alexa Fluor 568 (1:1000, A-11011, Thermo Fisher

Scientific, Waltham, MA), and Goat anti-Rabbit IgG (H+L) Alexa Fluor 633 (1:1000, A-21070, Thermo Fisher Scientific, Waltham, MA).

Confocal serial optical sections were obtained with a Fluoview FV3000 Confocal Microscope (Olympus, Tokyo, Japan).

For the Hr38 exonic probe, we first measured average fluorescence intensity within the cell boundary using Fiji (*Schindelin et al., 2012*; *Schneider et al., 2012*) and calculated the relative intensity to the background signals and then defined the positive cells whose signals were above 3σ. For each optical section, we defined as the background signal the area outside the region with clearly Hr38$^+$ cells, because the intensity of the background signal varies depending on the depth of the optical section.

For image registration, brain images were registered to T1 template brain (*Yu et al., 2010*) using the CMTK registration plugin of Fiji (https://flybrain.mrc-lmb.cam.ac.uk/dokuwiki/doku.php?id= warping_manual:registration_gui). To create average images for each condition, MATLAB was used to generate averaged images of each *Z*-plane from different individuals. For voxel-based heat map analysis of the *Hr38* expression, the median filter (pixel size = 10) in Fiji was applied to the averaged images. Fiji and Fluorender software (*Wan et al., 2009*) were used to create Z-stack images.

### catFISH

To prepare target flies, four to six particles of small iron filings (Delta Education, #060-0313 or equivalent) were attached on the dorsal side of the thorax of 6- to 8-day-old male or female flies using UV-activated glue under $CO_2$ anesthesia 1 day before the experiment. This allowed us to remove the target fly from a behavioral arena through a small hole using a neodymium magnet.

Six- to eight-day single-housed male flies were used for the experimental flies. A male fly was introduced in the behavioral arena and allowed to habituated to the new environment for ~3 hr. The first target fly was introduced into the arena, and the experimental fly was exposed to the target for 20 min. After the exposure, the target fly was removed from the arena as describe above. The experimental fly was given a 30-min interval before the second target was introduced. After 20-min exposure, the second target was removed, and *Hr38* expression was analyzed 10 min after the target fly removal (*Figure 4A, B*). Neurons were considered as positive for the Hr38 intronic probe when a clear dot signal with the intronic probe that co-localized with the Hr38 exonic signal was observed.

### Statistical analysis

Statistical analyses were performed using MATLAB (MathWorks, Natick, MA) and Python (Python Software Foundation). The detailed information including the number of samples, the statistical method, and the p value for each experiment is indicated in the figure legends.

### Calcium imaging data analysis

T-series images were manually screened and omitted from further analysis in cases where images were shifted in *Z*-direction, indicative of *Z*-motion artifacts. For image registration on the *X–Y* axis, T-series images were opened in Fiji by importing as Image Sequences. The Template Matching Plugin was used for motion correction. Briefly, from the Plugins menu, we selected Template Matching, then selected Align Slices in Stack. We then selected Normalized Correlation Coefficient as a Matching method and left other parameters as default values. We defined a landmark region containing the regions of interest (ROIs) by clicking and dragging to draw a rectangle, and initiated the registration process. After image registration, ROIs corresponding to nerve bundle or cell bodies of labeled neurons were manually selected and the fluorescence intensity in the ROIs was measured using Fiji. The average fluorescence intensity during the first 30 s (for neurites) or 15 s (for cell bodies) of image acquisition was used as the baseline to calculate the Δ*F/F*. The mean of Δ*F/F* during the indicated periods was calculated and visualized using Python. For cell body analysis, we defined the positive response if the average Δ*F/F* during the photostimulation was statistically higher than the baseline (p < 0.01). The strong response neurons (Δ*F/F* > 0.4) and the intermediate neurons (0.15 < Δ*F/F* ≤ 0.4) were defined based on the average Δ*F/F* value during the stimulation, respectively.

### Acknowledgements

We thank NA Pierce and M Schwarzkopf for advice on HCR techniques; A Sanchez for fly stock maintenance; S Cao for advice on imaging experiments; B Weissbourd for advice on catFISH experiments;

GM Rubin, BD Pfeiffer, and S Waddell for reagents; C Chiu, G Mancuso, and L Chavarria for the laboratory management and administrative assistance; and members of the Anderson laboratory for valuable comments on this work. This work was supported by NIDA Grant R01 DA031389. DJA is an Investigator of the Howard Hughes Medical Institute.

## Additional information

### Funding

| Funder | Grant reference number | Author |
|---|---|---|
| National Institute on Drug Abuse | DA031389 | David J Anderson |
| Howard Hughes Medical Institute | | David J Anderson |

The funders had no role in study design, data collection, and interpretation, or the decision to submit the work for publication.

### Author contributions

Kiichi Watanabe, Conceptualization, Resources, Data curation, Software, Formal analysis, Investigation, Visualization, Methodology, Writing – original draft, Writing – review and editing; Hui Chiu, Resources; David J Anderson, Conceptualization, Supervision, Funding acquisition, Methodology, Writing – original draft, Project administration, Writing – review and editing

### Author ORCIDs

Kiichi Watanabe ⓘ http://orcid.org/0000-0001-6505-8535
Hui Chiu ⓘ https://orcid.org/0000-0002-1820-8411
David J Anderson ⓘ https://orcid.org/0000-0001-6175-3872

Reviewer #1 (Public review): https://doi.org/10.7554/eLife.92380.3.sa1
Reviewer #2 (Public review): https://doi.org/10.7554/eLife.92380.3.sa2
Author response https://doi.org/10.7554/eLife.92380.3.sa3

## Additional files

### Supplementary files

• Supplementary file 1. The table provides information about the fly genotypes used in this research, including the names used in the manuscript, corresponding figure numbers, fly strain names, sources, identifiers, and FlyBase IDs.

• MDAR checklist

### Data availability

The data and code used for analyses in this paper are available in the GitHub repository (https://github.com/kiichiwatanabelab/Hr38_eLife, copy archived at *Watanabe, 2024*), and Dryad data repository (https://doi.org/10.5061/dryad.dr7sqvb7d). Any additional resources and reagents are available from the corresponding author upon request.

The following dataset was generated:

| Author(s) | Year | Dataset title | Dataset URL | Database and Identifier |
|---|---|---|---|---|
| Watanabe et al | 2024 | Data from: Whole brain in situ mapping of neuronal activation in *Drosophila* during social behaviors and optogenetic stimulation | https://doi.org/10.5061/dryad.dr7sqvb7d | Dryad Digital Repository, 10.5061/dryad.dr7sqvb7d |

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

# Appendix 1

## Supplementary protocol

HI-FISH with *Hr38* probe (from sample preparation)

### Sample preparation and fixation

Reagents

Ethanol

PBS

Fixative: 4% PFA/PBS (prepare on the day of the experiment)

1. Habituating flies in the arena for the experiments.

As mechanical stimulation during transfer induces *Hr38* expression, transfer flies to the arena for the experiments ~3 hr before the experiment. It is possible to extend this period to overnight, but in that case food should be included in the arena.

2. Perform photostimulation or behavioral assay to induce *Hr38* expression.

3. For simple *Hr38* detection with the exonic probe, wait ~30 min before beginning fixation and dissection.

4. Prefix the samples.

(This procedure can be omitted if there is a single brain to be dissected).

For dissecting multiple brains, prefix the samples before dissection.

Collect individual flies, soak them in cold ethanol to remove the wax. Transfer them into cold PBS. Remove proboscis and open head cuticle quickly.

Transfer it to the cold fixative in a dish on ice (using a multi-well glass dish). After completing this procedure for all individual flies, keep the dish with the fixative on the ice.

Transfer one fly to the dissection dish (another multi-well glass dish) filled with ice-cold PBS and keep the samples on ice.

5. Dissect fly brains as quickly as possible.

Transfer one fly to another dissection dish (multi-well glass dish) filled with PBS (room temperature).

Dissect a brain and transfer it to the tube with fixative on ice. Dissection procedure is performed under a dissection microscope at room temperature.

6. Fixation (rotating/shaking, at 4°C for at least 2 hr ~O/N).

### Immunostaining (if desired)

(In cases where HCR3.0 in situ is combined with immunostaining, perform the immunostaining first under RNase-free condition, post-fix with 4% PFA for 20 min and then proceed to the following 'Pre-treatment' procedure. Otherwise, proceed to 'Pre-treatment' after 'Sample preparation and fixation'.)

Perform regular immunostaining with fluorescent labeled antibodies on fly brains. For immunostaining under RNase-free condition, use nuclease-free reagents.

Followings are some examples of these reagents and their suppliers.

| | |
|---|---|
| Nuclease-free water | Thermo Fisher AM9932 |
| Phosphate-buffered saline (10×) pH 7.4, RNase-free | Thermo Fisher AM9624 |

For reagents or buffers that are difficult to be prepared while maintaining RNase-free conditions (such as the antibody solution from DSHB hybridoma supernatant), RNasin Plus Ribonuclease Inhibitor is added (1/50 vol. Promega N2611).

### Pre-treatment

Reagents (RNase free)

PBS

PBS 0.1% Triton X-100

PBS 0.5% Tween-20

2% acetic acid (ice-cold)

1. 4% PFAWash samples 3 × 5 min with PBS/0.1% Triton X-100 at room temperature to remove the fixative.

2. Permeabilization with PBS/0.5% Tween-20 at room temperature for 20 min.
3. Rinse with PBS × 2.
4. Treat samples with ice-cold 2% acetic acid for 1 min (*).
5. Rinse with PBS × 2.
6. Post-fix with 4% PFA for 20 min on ice.
7. Wash samples 3 × 5 min with PBS/0.1% Triton X-100 at room temperature.
8. Proceed to HCR v3.0 procedure.

\* For most of applications targeting cytoplasmic mRNA detection, acid treatment can be omitted. For detecting nuclear signals, acid pre-treatment improves HCR signals (particularly, the signal of the Hr38 intronic probe), but higher concentration/longer treatment decreases fluorescence signals (either native GFP fluorescence or Immunocytochemistry). Optimize the condition according to the type of labeling performed in each experiment. In general, 2% acetic acid for 1 min is acceptable.

## HCR3.0

Follow the protocol from Molecular Technologies with minor modifications
(https://files.molecularinstruments.com/MI-Protocol-RNAFISH-GenericSolution-Rev8.pdf)
Use the following hybridization buffer for better preservation of tissue morphology (simply replace Formamide in the original protocol with 8 M urea).
(ref. https://doi.org/10.1016/j.ydbio.2017.11.015)

30% urea (8 M): final 2.4 M
5× SSC
9 mM citric acid (pH 6.0)
0.1% Tween-20
50 µg/ml heparin
1× Denhardt's solution
10% dextran sulfate

Ordering information for modified hybridization buffer

| | |
|---|---|
| 8 M urea | Merck U4883 |
| 20× SSC | Thermo Fisher AM9763 |
| Citric acid monohydrate | Merck C0706 |
| 50% Tween-20 | Thermo Fisher 3005 |
| Heparin sodium salt | Merck H3393 |
| Denhard's solution (50×) | Thermo Fisher 750018 |
| Dextran sulfate 50% solution | Merck S4030 |

Hybridization reactions are performed in a 1.5-ml Eppendorf tube (nuclease-free).
~8 brains in ~50 µl hybridization solution (scaled down from the original protocol)

## Sample mounting

We prefer to use Histodenz mounting solution, adopted from the Gradinaru lab procedure.
doi: 10.1016/j.cell.2014.07.017

