## [Editor Report · eLife Assessment]

This work reports an **important** new method for activity-dependent neuronal labeling in *Drosophila* using in situ hybridization, with the potential to establish a new standard in the field. The authors demonstrate the method's applicability by generating **compelling** evidence of the function of male-specific neurons in both aggression and courtship behaviors. These results and the new method will be of great interest to the neuroscience community.

---

## [Referee Report · Reviewer #1 (Public review)]

Summary:

The authors have nicely demonstrated the efficiency of the HCR v.3.0 using hr38 mRNA expression as a marker of neuronal activity. This is very important in the *Drosophila* neuroscience field as in situ hybridization in adult *Drosophila* brains have been so far very challenging to do and replicate. The HCR v.3.0 has been described before [Choi et al., (2018)] and is now the property of the non-profit organization Molecular Technologies, who are the ones responsible for designing the probes. Here, taking advantage of this new FISH method, the authors have demonstrated the use of the FISH to identify neurons activated by a specific behavioral task using hr38 mRNA as a marker of neuronal activation. They named their method HI-FISH.

In addition, based on the catFISH method [Guzowski et al., 1999], the authors were able to distinguish between newly activated neurons (nascent nuclear mRNA) and mature hr38 mRNA showing an earlier activation. They describe this method as HI-catFISH.

Finally, to test what are the neurons activated downstream of their neuronal group of interest, the authors combined the HI-FISH method with optogenetic using chrimson. They named this method opto-HI-FISH.

Using these three new methods, the authors have addressed the following biological question: are love and aggressiveness neuronally the same in *Drosophila*?

Here, the authors focused on the male specific P1a neurons which are activated by both an aggressive context (male-male encounter) and sexual context (male female encounter).

Strengths:

The demonstration of the efficiency of the method is very convincing and well-performed. It gives the will for the reader to apply the method to their own subject.

Weaknesses:

The more neurons are present, the more difficult it is to identify neurons. This is something to take into account when applying these methods.

---

## [Referee Report · Reviewer #2 (Public review)]

Summary:

Watanabe et al. introduce a novel approach for activity-dependent labeling of neural circuits in *Drosophila* at single-cell resolution, based on detecting the expression of the immediate early gene Hr38 using in situ hybridization. While activity mapping of neurons during specific behaviors is well-established in rodent models, its application in *Drosophila* has been limited, primarily due to technical constraints. By overcoming these challenges, this study tackles an important and timely issue, providing a foundational tool that will serve as a key reference in the field of circuit neuroscience.

Strengths:

The principal strength of this method lies in its versatility and high sensitivity. It can be applied to a broad range of biological questions and enables the investigation of dynamic transcriptional regulation across an unlimited number of genes with a strong signal-to-noise ratio. As such, it holds great potential for widespread use across research labs.

Weaknesses:

No major weaknesses; all concerns have been adequately addressed.

---

## [Author Response]

**Reviewer #1:**

Response to Public Review

We thank the reviewer for taking the time to carefully read our paper and to provide helpful comments and suggestions, most of which we have incorporated in our revised manuscript. One of this reviewer’s (and reviewer #2’s) main concerns was that the confocal images provided in some cases did not appear to reflect the quantitative data in the bar graphs. These images were provided only for illustrative purposes, to give the reader a sense of what the primary data look like. The reviewer may not have appreciated that the quantitative data reflect counts of RNA smFISH signals (dots) in hundreds of cells collected through z-stacks comprising multiple optical sections in multiple flies for each condition For example, in P1a control condition (in Figure 2A), we have analyzed 135 neurons from 8 individuals. There, the number of z-planes ranged from 3 to 8 per hemisphere. It is generally not possible to find a single confocal section that encompasses quantitatively the statistics that are presented in the graphs. Presenting the data as an MIP (Maximum Intensity Projection, i.e., collapsed z-stack) in a single panel would generate an image that is too cluttered to see any detail. We have now included, for the reader’s benefit, additional example confocal sections in both a z-stack and from the opposite hemisphere, in Supplemental Figure S4D. We have also inserted clarifying statements in the text on p. 7 (lines 154-156).

Another suggestion from Reviewer #1 is that "it would be more informative to separate in the quantification between the GAL4-expressing neurons and the non-expressing ones" based on the presented pictures where more non-P1a neurons (that the reviewer speculates may be pC1-type neurons) are activated by a male-male encounter than by a male-female encounter, while the P1a-positive neurons seem to be more responsive during courtship behavior. In this paper, we were not looking at pC1 neurons and did not try to answer which neuronal population(s) outside of the P1a population is/are responsible for aggression and/or courtship. Rather, we focused on P1a neurons and addressed whether P1a neurons that induce both aggression and courtship behavior when they are artificially activated (Hoopfer et al. 2015) are also naturally activated during spontaneous performance of these two social behaviors. However, this result did not exclude the possibility that P1a neurons were inactive during naturalistic courtship or aggression. Our data in the current manuscript provide further experimental evidence in support of the idea that P1a neurons as a population play a role in both of these behaviors. Moreover, we provided data identifying P1a neurons activated only during aggression or during courtship (or both). However this does not exclude that pC1 or other neighboring populations are activated during aggression as well (See also the response to 'Recommendations For The Authors' and text lines 151-154).

In Figure 3, we used opto-HI-FISH to identify candidate downstream targets (direct or indirect) of P1a neurons. We used 50 Hz Chrimson stimulation to activate P1a neurons to induce expression of Hr38 and identified Kenyon cells in the mushroom body (MB) and PAM neurons (as well as pCd neurons) as potential downstream targets of P1a cells. In Figure 3 – supplement we performed calcium imaging of KCs and PAM neurons in response to P1a optogenetic stimulation to confirm independently our results from the Hr38 labeling experiments. That control was the purpose of that supplemental experiment.

Based on those imaging data, the reviewer asked the further question of which [natural] behavioral context induces Hr38 expression in these populations (i.e., mating or aggression). This question is reasonable because our calcium imaging data (Figure 3-supplement) showed that both Kenyon cells and PAM neurons are active only during photo-stimulation of P1a neurons. Our previous behavioral studies (Inagaki et al., 2014; Hoopfer et al., 2015) showed that 50 Hz photo-stimulation of P1a neurons in freely moving flies induced unilateral wing extension during stimulation, while aggression was observed only after the offset of the stimulation (Hoopfer et.al., 2015). Based on the comparison of those behavioral data to the imaging results in this paper, the reviewer suggested that Kenyon cells and PAM neurons are activated during courtship rather than during aggression. This is certainly a possible interpretation. However it is difficult to extrapolate from behavioral experiments in freely moving animals to calcium imaging results in head-fixed flies, particularly with response to neural dynamics. Furthermore, Hr38 expression, like that of other IEGs (e.g., c-fos), may reflect persistently activated 2nd messenger pathways (e.g., cAMP, IP3) in Kenyon cells and PAM neurons that are not detected by calcium imaging, but that nevertheless play a role in mediating its behavioral effects. We still do not understand the mechanisms of how optogenetic stimulation of P1a neurons in freely behaving flies induces aggression vs. courtship behavior. Although 50 Hz stimulation of P1a neurons does not induce aggressive behavior during photo-stimulation, it is possible that this manipulation activates both aggression and courtship circuits, but that the courtship circuit might inhibit aggressive behavior at a site downstream of the MB (e.g., in the VNC). Once stimulation is terminated and courtship stops the fly would show aggressive behavior, due to release of that downstream inhibition (see Models in Anderson (2016) Fig 2d, e). In that case, there would be no apparent inconsistency between the imaging data and behavioral data. We agree that the reviewer's question is interesting and important but we feel that answering this question with decisive experiments is beyond the scope of this manuscript.

Finally, Reviewer #1 suggested a method to evaluate the Hr38 signals in the catFISH experiment of Figure 4. We appreciate their suggestions, but the way that we evaluated the Hr38 signals was basically the same as the way the reviewer suggested. We apologize for the confusion caused by the lack of detailed descriptions in the original manuscript. We have now revised the methods section to explain more clearly how we define the cells as positive based on Hr38EXN and Hr38INT signals.

Response to Recommendations for the authors:

“To strengthen the author's argumentation, I would distinguish in their quantification between gal4+ from the other [classes of neighboring neurons]” (Fig. 2 and 4).”

Our focus in this paper was to ask simply whether P1a neurons are active or not active during natural occurrences of the social behaviors they can evoke when artificially activated. We did not claim that they are the only cells in the region that control the behaviors. It is not possible to compare their activation to that of 'other' cells neighboring P1a neurons without a separate marker to identify those cells driven by a different reporter system (e.g., LexA). This in turn would require repeating all of the experiments in Figs 2 and 4 from scratch with new genotypes permitting dual-labeling of the two populations by different XFPs, and quantifying the data using 4-color labeling. We respectfully submit that such curiosity-driven experiments, while in principle interesting, are beyond the scope of the present manuscript. However, we have inserted text to acknowledge the possibility that the aggression-activated Hr38 signals in P1a- cells neighboring P1a+ cells may correspond to other classes of P1 neurons (of which there are 70 in total) or to pC1 cells. Changes: Text lines 151-154.

“if the magenta dot is outside of the nuclei I would not count this as positive also the size of the dot seems to be a good marker of the reality of the signal. I would measure the intensity of the hr38EXN. A high Hr38EXN level associated with the presence of hr38INT would indicate that the cell has been activated during both encounters, while a lower hr38EXN with no hr38INT would suggest only an activation during the 1st behavioural context. Finally, a lower hr38EXN associated with the presence of hr38INT would suggest the opposite, an activation only during the 2nd behaviour.”

We agree that there are some tiny dot signals with hr38 INT probe that are more likely the background signals. We only counted the INT probe signals as positive when the cells had a clearly visible dot and also co-localize with the exonic probe's signal, as primary (un-spliced) Hr38 transcripts in the nucleus should be positive for both EXN and INT probes. Regarding the reviewer’s latter comments, we agree with their interpretation of the catFISH results and that is how we interpreted them originally. We measured the intensity of hr38EXN expression and defined hr38EXN-labeled cells as “positive” when the relative intensity was 3σ >average, a stringent criterion. In the revised manuscript, we added more detailed information in the methods section regarding our criteria for defining cell types as positive.

“Knowing that the P1a neurons (using the split-gal4) can trigger only wing extension when activated by optogenetic 50Hz, I would test to which behavioral context the MB neurons and the PAM neurons positively respond to.”

As we answered in 'Response to Public Review,' our opto-HI-FISH experiments identified Kenyon cells in the mushroom body (MB) and PAM neurons (as well as pCd neurons) as potential downstream targets of P1a cells, using Hr38 labeling. The purpose of the calcium imaging experiment in Figure 3 – supplement was to confirm the P1a-dependent activation of KCs and PAM neurons using an independent method. In that respect this control experiment was successful in that methodological confirmation. The reviser raised an interesting question about how our calcium imaging experiments relate to our behavioral experiments, in terms of the dynamics of KC and PAM activation. A recent publication (Shen et al., 2023) revealed that courtship behavior has a positive valence and that activation of P1 neurons mimics a courtship-reward state via activation of PAM dopaminergic neurons. Therefore, it is reasonable to think that PAM neurons (and Kenyon cells as downstream of PAM neurons) are activated during female exposure. However those data do not exclude the possibility that inter-male aggression is also rewarding in *Drosophila* males, as it has shown to be in mice. This is an interesting curiosity-driven question that has yet to be resolved. Therefore, as mentioned in the 'Response to Public Review,' we feel that the additional experiment the reviewer suggests is beyond the scope of our manuscript.

Changes: None.

Minor comments:“Please provide different pictures from main fig2 and sup2 for the three common conditions (control, aggression, and courtship).”

The data set for Figure 2 and Figure 2 supplement are from the same experiment. Because of the limited space, we just presented the selected key conditions ('Control', 'Aggression', and 'Courtship') in the main figure and put the complete data set (including these three key conditions) in the supplemental figure.

Changes: None

“Please, provide scale bars for the images.”Also, Reviewer #2 commented, 'Scale bars are missing on all the images throughout the main and supplementary figures.'

We have now added scale bars for each figure.

“Fig.1: “Is the chrimsonTdtom images from endogenous fluorescence? It is not said in the legend and anti-dsred is not provided in the material and method while anti-GFP is.”

We are sorry for the confusion and thank the reviewer for raising that question. The signals were native fluorescence, and we have now added that information to the figure legend.

P7: "As an initial proof-of-concept application of HI-FISH, we asked whether neuronal subsets initially identified in functional screens for aggression-promoting neurons (Asahina et al., 2014; Hoopfer et al., 2015; Watanabe et al., 2017) were actually active during natural aggressive behavior. These included P1a, Tachykinin-FruM+ (TkFruM), and aSP2 neurons". Please put the references to the corresponding group of neurons listed. For example: "These included P1a neurons [Hoopfer et al., 2015]".

We have now added these references.

P9: "Optogenetic and thermogenetic stimulation experiments have shown that that P1a interneurons can promote both male-directed aggression and male- or female-directed courtship" typo

We appreciate the reviewer for catching this error and have corrected the text.

P10:" To validate this approach, we first asked whether we could detect Hr38 induction in pCd neurons, which were previously shown by calcium imaging to be (indirect) targets of P1a neurons". Reference [Jung et al., 2020]

We have now added this reference.

Fig. 4A: Put the time scale on the diagram (3h adaptation-20min-30min rest-20min-10min rest-collect)

We have now added the time scale in Figure 4A.

**Reviewer #2:**

Response to Public Review:

We thank the reviewer for their helpful comments and suggestions. We have addressed most of them in our revised manuscript. The main concern of Reviewer #2 was the temporal resolution of the HI-catFISH experiment shown in Figure 4 and Figure 4-Supplement. Our original manuscript illustrated temporal patterns of Hr38EXN and Hr38ITN signals concomitant with different behavioral paradigms (Figure 4B). The reviewer pointed out that the illustrated experimental design does not reflect the actual data shown in Figure 4-Supplement A-C. We believe this issue was raised because we drew the temporal pattern of Hr38EXN signals in Figure 4B based on the intensity of Hr38EXN signals (Figure 4-Supplement B) rather than based on the % number of positive cells (Figure 4-Supplement C). We have now revised the schematic time course of Hr38EXN signals in Figure 4B using the % of positive cells. We believe this change will be helpful for readers to understand better the experimental design since we used the % of positive cells to identify patterns of P1a neuron activation during male-male vs. male-female social interactions in Figure 4D. Another suggestion from Reviewer #2 was to add additional controls, such as the quantification of the intronic and exonic Hr38 probes after either only the first or second social context exposure. In response, we have now added the data from only the first social context (Figure 4C, and 4D, right column). These new data provides evidence that there are essentially no detectable Hr38INT signals 60 minutes later without a second behavioral context, while Hr38EXN signals are still present at the time of the analysis. Unfortunately, we are not able to provide the converse dataset with the second behavioral context only to show that Hr38 INT signals are detected. On this point, we call the reviewer’s attention to Figure 4-supplement-S4A-C, which show that the INT probe signals are detectable at 15 and 30 minutes following stimulation, but not at 60 minutes. In the experiment of Fig. 4B, flies are fixed and labeled for Hr38 30 minutes after the beginning of the second behavior, conditions under which we should obtain robust INT signals (as observed). EXN signals are also expected at 30 minutes because the primary (non-spliced) RNA transcript detected by the INT probe also contains exonic sequences.

Response to Recommendations for the authors:

Given that the development of in situ HCR for the adult fly brain is so central to the present manuscript, I think that the methods section describing the HCR protocol can be significantly improved. In particular, the authors should fully describe the in situ HCR protocol including the 'minor modifications' they refer to, and define how they calculate the 'relative intensity to the background'.

We appreciate the reviewer’s suggestion. We have now revised the methods section to describe the procedure in more detail. Also, we will submit a separate document describing the HI-FISH protocol.

Note: The authors refer to a recently published paper by Takayanagi-Kiya et al (2023) describing activity-based neuronal labeling using a different immediate early gene, stripe/egr-1. The authors state the following: 'That study used a GAL4 driver for the stripe/egr-1 gene to label and functionally manipulate activated neurons. In contrast, our approach is based purely on detecting expression of the IEG mRNA using..'. Takayanagi-Kiya et al. (2023) also use in situ mRNA detection of the IEG stripe/egr-1 and not only a GAL4 driver system. This claim should be modified and the paper should be cited in the introduction of the present paper.

We have now cited the paper in the Introduction and have modified and moved the description originally in 'Note' section to Discussion (text lines: 392-404) as the reviewer requested. We have emphasized the difference between the two approaches for comparing neuronal activities during two different behaviors within the same animal. Takayanagi-Kiya used GAL4/UAS and stripe protein expression with immunohistochemistry to analyze neuronal activities during two different behaviors, while we exclusively analyzed Hr38 mRNA expression for this purpose, using intronic and exonic Hr38 probes. This approach made it possible to perform catFISH with higher temporal resolution and also allows extension of our approach to other IEGs for which antibodies are not available.

Please specify the nature of the iron fillings in the methods section.

We added a detailed description in the methods section, including the catalog number.

In Figure 1B, the authors may add a dashed outline to the regions magnified in 1C so that readers can more easily follow the figures. Moreover, it would be informative to see a more detailed quantification of the number of Hr38-positive cells in different brain regions marked by Fru-GAL4.

We have now added the whole brain images for each condition in Figure 1C and also quantitative data in Figure 1-Supplement C, as the reviewer suggested.

In the middle right aggression panel of Figure 2A, it looks as if one P1a neuron is not outlined.

We have carefully examined other z-planes through this region and based on those data have concluded that the signals mentioned by the reviewer are neurites from neurons labeled in other z-planes.

Changes: None.

The images in Figure 2A can be again found in Figure Supplement 2A, yet the number of neurons analyzed suggests the quantification was performed from different samples. The images in Figure Supplement 2A should be either changed or it should be explained as to why the images are the same yet the numbers in the legend are different.

We apologize for the confusion. Figure 2 and Figure 2-Supplement are from the same experiment. To avoid clutter we illustrated three key conditions ('Control,' 'Aggression,' and 'Courtship') in the main figure. The reason why the numbers in the legend are different is that the purpose of presenting Figure 2-Supplement B-D was to determine whether there were differences in the intensity of Hr38 FISH signals in the neurons considered as 'positive' in different conditions. Therefore, the numbers described in Figure 2-Supplement legend are derived only from those neurons that were considered Hr38-positive, while the numbers in Figure 2 include all neurons analyzed. We have now added notes to explain this in the Figure 2 – supplement legend.

The panels of the quantification of the Hr38 relative intensity in Figure 2B/C/D are very difficult to read, ideally, they should be plotted as in Figure Supplement 2B/C/D.

The graphs in Figure 2B-D (upper) show data from all GFP-labeled cells scored, including cells defined as 'negative' or 'borderline.' In contrast, the graphs in Figure 2-supplement show the relative Hr38 signal intensity in those GFP neurons defined as positive based on the analysis in Fig. 2B. If we were to plot the data in Fig. 2B (upper) as box plots (like that in Figure-2-supplement), we would see either a skewed (only negative cells) or a bimodal distribution (one around the negative population and the other around the positive population); the shapes of these distributions would likely be hidden in the box-whisker plots format. Therefore, we prefer to plot all of the data points as we did in the original manuscript. However, we agree that the data points in the original manuscript were hard to read. We therefore changed the format of the datapoints from blurry dots to open circles with clear solid lines.

In Figure 2B/C/D, please specify in the figure legend what 'grouped in categories according to character' means.

We used letters to mark statistically significant differences (or lack thereof) between conditions. Bars sharing at least one common letter are not significantly different. If they do not share any letter, they are significantly different. For example, Aggression: bc vs. Dead: bc, means no difference. Aggression: bc vs. No Food: b, or Aggression: bc vs. Courtship: c also means no difference between Aggression and each of the two other conditions. However, 'No Food: b' and 'Courtship: c' have no common letter, meaning they are different. This is a standard method for showing statistically comparisons among multiple bars without lots of asterisks and horizontal bars cluttering the figure, and we have revised the legend to clarify what each letter means. We have also removed the color shading in Figure 2 B-D as it may have been confusing.

A quantification of the number of Hr38-positive neurons and Hr38 relative intensity during the entire time course would be informative in Figure 3D.

Although the data set for this figure is different from that for Figure 4-Supplement A-C, the main claim is the same. Therefore, Figure 4 - Supplement essentially provides the information that the reviewer suggested. However, we also reanalyzed the data set used for the original Figure 3D and evaluated % positive cells at the 30-minute time point and have now added that number in the figure legend.

In the legend of Figure 3D, it says '..The expression level reaches its peak at 30-60min', yet I don't see timepoints beyond 60min. Please rephrase or add additional timepoints.

We apologize for the error. We have rephrased the text.

Figure Supplement 3A/D: please add an outline or a schematic figure to better understand where the imaging is performed.

We added illustrated schemas next to the title of each experiment (P1->PAM neurons (bundle) and P1 -> Kenyon cells (bundle)).

Figure Supplement 3C/F: please add information about the statistical test to the corresponding figure legend.

We have added a phrase to describe the test used.

Figure Supplement 3G/H/I/J: motion artifacts can potentially strongly affect the performed analysis given that cell bodies are very small and highly subjected to motion. Can the authors comment on how they corrected for motion?

We have now described how we corrected for motion artifacts in the Methods section.

Figure 4C/D: It seems as if the representative images don't reflect the quantification, e.g., in the male -> female panel, close to 100% of the neurons are positive for the exonic probe as opposed to approx. 40% in the bar graph.

Please see our response to this issue in the 'Response to Public Review (Reviewer #1)'.

Additional controls should be included in Figure 4C in order to assess the temporal resolution of HI-CatFISH more in detail (see 'Weaknesses').

We have also answered this in the 'Response to Public Review'.

The authors should adjust the scheme in the main Figure 4B to reflect the data presented in Figure S4A and C. For instance, the peak for the intronic version is observed at 15 minutes, while at 30 minutes, both the exonic and intronic signals show an equal level of signal.

We have addressed this issue in the 'Response to Public Review'.

We thank the reviewers again for their helpful comments and hope that with these changes, the manuscript will now be acceptable for official publication in eLife.